# Breakfast and Exercise Improve Academic and Cognitive Performance in Adolescents

**DOI:** 10.3390/nu13041278

**Published:** 2021-04-13

**Authors:** Masato Kawabata, Kerry Lee, Hui-Cheng Choo, Stephen F. Burns

**Affiliations:** 1Physical Education and Sports Science, National Institute of Education, Nanyang Technological University, Singapore S637616, Singapore; masato.kawabata@nie.edu.sg (M.K.); zu_warriors@yahoo.com (H.-C.C.); 2Department of Early Childhood Education, The Education University of Hong Kong, New Territories, Hong Kong, China; kerrylee@eduhk.hk

**Keywords:** cognition, breakfast, glycemic index, physical activity

## Abstract

This study examined the combined effects of breakfast and exercise on short-term academic and cognitive performance in adolescents. Eighty-two adolescents (64 female), aged 14–19 years, were randomized to four groups over a 4-hour morning: (i) a group who fasted and were sedentary (F-S); (ii) a group who ate breakfast but were sedentary (B-S); (iii) a group who fasted but completed a 30-min exercise bout (F-E); and (iv) a group who ate breakfast and completed a 30-min exercise bout (B-E). Individuals completed academic and cognitive tests over the morning. Adolescents in B-E significantly improved their mathematics score (B-E: 15.2% improvement on correct answers, vs. F-S: 6.7% improvement on correct answers; *p* = 0.014) and computation time for correct answers (B-E: 16.7% improvement, vs. F-S: 7.4% improvement; *p* = 0.004) over the morning compared with the F-S group. The B-E group had faster reaction times for congruent, incongruent and control trials of the Stroop Color-Word Task compared with F-S mid-morning (all *p* < 0.05). Morning breakfast and exercise combine to improve short-term mathematical task performance and speed in adolescents.

## 1. Introduction

Good evidence demonstrates the benefits of regular breakfast intake on cognitive function in children and adolescents [1]. Much of the influence of breakfast on cognition likely results from acute intake of food rather than a long-term improvement in nutrient status [1,2,3,4,5]. The immediate benefits of breakfast are evident on measures of memory and in terms of fewer errors on tasks related to attention, especially later in the morning when performance deteriorates with fasting [1,3,5]. Satiation and hunger alleviation, improved fuel and nutrient provision for the central nervous system, and an increase in the synthesis or levels of brain neurotransmitters are all mechanisms via which breakfast may mediate short-term improvements in cognition [1,4,6,7]. Breakfast composition is one aspect of important consideration with lower glycemic index (GI) or glycemic load breakfasts appearing favorable in terms of cognitive outcomes [2,3,4,6,8]. This may result from better blood glucose regulation or, alternatively, from a favorable hormone or neurotransmitter response [1,6].

Whilst breakfast can improve acute cognitive function, effects on academic performance or attainment are more difficult to demonstrate. An early randomized controlled study in 405 undernourished and 405 well-nourished rural Jamaican children suggests small improvements in test achievement scores in those children who were given breakfast to consume [9]. More recently, observational data support the effect of regular breakfast intake on academic performance and grades and longitudinal data from the United Kingdom have shown an association of regular breakfast consumption with performance measured 6 and 18 months before Statutory Assessment Tests [10]. From the perspective of the acute intake of food, breakfast is unlikely to cause any lasting changes in academic outcomes. However, it seems logical to suggest that through its acute influence on various aspects of cognitive function, such as attention, this could translate into better performance in academic tests, although we are not aware of data showing this.

Regular physical activity also has a positive relationship with academic and cognitive outcomes [11,12,13,14]. Randomized controlled trials have found increased attentional inhibition and cognitive flexibility with improvements in fitness in children [15]. Improved executive function and mathematics achievement that show a dose–response relationship with exercise training [16] have also been demonstrated. However, as with breakfast, there are immediate benefits of exercise on cognition not resulting from changes in fitness. Speed of processing, response accuracy, concentration and improvements in mathematical computation can all improve after acute bouts of exercise; although the optimal type, amount, and frequency of exercise need better clarification [11,12,13,14]. One important note is that most studies on the relationship of physical activity with academic performance and cognitive outcomes are in younger children aged 6–13 years. A paucity of rigorous randomized designs in adolescents (aged 14–18 years) led the 2018 Physical Activity Guidelines Advisory Committee in the US [17] to conclude that there was insufficient evidence available to provide even a limited grade for the effect of physical activity on cognition in adolescents. A subsequent follow-up review in 2019 found “limited but promising evidence” for positive effects of physical activity on cognition in adolescents [13]. For academic outcomes, one recent meta-analysis found a positive effect (Cohen’s d = 0.37) of physical activity based on 10 studies [18]. For cognition, several systematic reviews have generally found a positive relationship with physical activity in adolescents but with some heterogeneity in the findings [19,20,21,22,23]. Thus, further research in this age group is justified.

One interesting observation in adolescents relates to acute coordinative exercise, which emphasizes the ability to balance, react, adjust and to differentiate within a short time. A study in German adolescents found that only 10 min of this type of exercise led to improvements in attention and concentration on a paper and pencil letter cancellation test compared with those who completed moderate intensity exercise without any specific coordinative request [24]. Potentially, coordinative exercise may activate parts of the cerebellum and areas of the prefrontal lobe related to mathematics processing and reading comprehension to a greater extent than other types of exercise [24]. Further substantiation of the benefits of this type of exercise on cognition in adolescents is required.

Although both breakfast and exercise independently improve cognitive function, we are aware of only one study examining how they interact in children [25]. In that investigation, 42 early adolescents (12.4 (0.4) years), were randomized to receive either a high or low GI breakfast and then complete two trials in a random order which involved either a morning of rest or exercise taken 1 h after breakfast. Cognitive tests 30 min before and 90 min after exercise showed that the low GI breakfast and exercise were individually beneficial for improving response times on a test of working memory but conferred additive benefits on response times for a test of selective attention and inhibition. Conversely, for the high GI breakfast response times for selective attention and inhibition improved only under conditions of rest whereas exercise alone improved response times of working memory. This demonstrates the importance of elucidating the combined effects of breakfast and exercise on cognition. Further work needs to examine if this can result in improved academic as well as cognitive outcomes.

Thus, the present investigation examined the combined effects of exercise and breakfast intake on academic and cognitive outcomes over a morning in adolescents. We combined a low GI breakfast with a mixture of 30 min of aerobic and bilateral coordinative exercise. We hypothesized that adolescents in the combined exercise and breakfast intervention would show better outcomes than those who remained sedentary and fasted over a morning or adolescents given only exercise or breakfast.

## 2. Materials and Methods

### 2.1. Participants

This study was conducted according to the guidelines laid down in The Code of Ethics of the World Medical Association in the Declaration of Helsinki 1964 and its subsequent amendments. All procedures involving human subjects were approved by Nanyang Technological University Institutional Review Board (IRB-2014-01-030). Adolescents were recruited from four local junior colleges and high schools with the consent of the Ministry of Education Singapore and the Principle of each institute. Written informed parental consent and adolescent assent was obtained from all subjects involved in the study before testing began. Inclusion criteria for the study were: (i) age: 14–19 years; (ii) body mass index normal for age and sex as defined by the Ministry of Health in Singapore [26]; (iii) no clinically diagnosed learning or attention disorders; (iv) no clinically diagnosed eating disorders; (v) free of diabetes mellitus; (vi) not partaking in any dietary restrictions for personal or religious beliefs (e.g., Ramadan); (vii) exercise training < 5 days per week for < 20 min each session (determined by self-report) and (viii) not color blind (one cognitive test used color coding).

### 2.2. Anthropometric and Preliminary Testing

Participants underwent anthropometric measures of stature, body mass and waist circumference. Stature was measured to the nearest 0.1 cm using an electronic wall-mounted stadiometer (Seca 242, Seca, Hamburg, Germany). Body mass was recorded to the nearest 0.1 kg using an electronic scale (Mettler-Toledo ID1 Plus, Mettler-Toledo S.E.A Pte Ltd., Singapore). Waist circumference was measured to the nearest 0.1 cm using a flexible measuring tape at the midpoint between the lowest rib and the iliac crest. Resting blood pressure and pulse rate were measured in duplicate using an automated sphygmomanometer (Omron IW2, Omron Healthcare, Kyoto, Japan). Following these measures, all individuals completed a preliminary 20 m multi-stage shuttle run test to assess cardiorespiratory fitness [27]. Heart rate was recorded continuously throughout the tests and used to predict peak oxygen uptake from standardized regression equations [27] and to determine running speed for the subsequent main trials.

### 2.3. Main Trials

Participants were randomized to one of four groups for the main trials: (i) a group who were fasted and sedentary all morning (F-S); (ii) a group who ate breakfast but were sedentary all morning (B-S); (iii) a group who fasted all morning but completed a 30 min exercise bout (F-E); and (iv) a group who ate breakfast and completed a 30 min exercise bout (B-E). Participants were transported to the laboratory by 08:30 a.m. by car, taxi or public transport having not eaten food or consumed any liquids other than water ad libitum since 23:00 p.m. the previous evening.

Upon arrival in the laboratory, participants sat in a chair for 10 min before a fingertip blood sample was taken for a baseline measure of blood glucose. Ten minutes later, participants were provided a series of baseline academic and cognitive tests (test-series_1_). Motivation was assessed before (pre-test_1_) the tests and hunger, satiety, fullness and appetite, arousal, and feeling before and after (pre-test_2_) the test-series in the same order. Mental effort was assessed immediately after the test-series (pre-test_2_). The entire set of procedures took ~30 min to complete. At 09:30 a.m., participants in the F-E and B-E groups performed a single 30 min bout of exercise consisting of treadmill running and bilateral coordinative ball exercises. Participants in the F-S and B-S groups engaged in quiet activities during this time (allowed to use computers/phones, work or read but were not allowed to engage in gaming activities). At 10:00 a.m., participants in the B-S and B-E groups were given a standardized breakfast to consume within 15 min while those in the F-S and F-E groups rested. A second battery of cognitive tests only (test-series_2_) was provided at 10:20 a.m. with the same set of subjective measures given for test-series_1_ provided (post-test_1_) and after (post-test_2_). Participants then remained seated engaging in quiet activities for a further 60 min. A final battery of academic and cognitive tests (test-series_3_) was then provided to participants along with the same subjective measures before (post-test_3_) and after (post-test_4_). Immediately after these final measures, a second fingertip blood sample for the measurement of blood glucose was taken. Participants were unable to eat anything all morning except for the breakfast provided but were allowed to drink water ad libitum. A schematic of the study design over the morning is provided in Figure 1.

### 2.4. Exercise

Participants in the two exercise trials, F-E and B-E, completed a 30 min bout of exercise between the first and second test-series. The exercise itself involved an 8 min run on a treadmill at a speed equivalent to 60% of peak oxygen uptake. Heart rate was recorded continuously during running and rating of perceived exertion (RPE) [28] at the end of the run. This was followed by 20 min of bilateral coordinative ball exercises adapted from a published protocol based on coordinative training forms for soccer and exercises from the Munich Fitness Test [24]. These exercises stressed different bilateral coordinative abilities within a short amount of time and were organized into five stations with participants completing two rounds of 90 s work at each station and 30 s rest between stations.

At station 1, participants bounced a basketball alternating with the left or right hand while standing on two wobble boards. For station 2, the task was to bounce a basketball and volleyball with the left and the right hands simultaneously. The balls were swapped between hands midway through each 90 s bout. At station 3, the task was to throw a handball alternating with the left and right hand into a gymnastic hoop held at waist level by an experimenter at a distance of 5 m for girls and 8 m for boys. In station 4, participants faced an experimenter, who served as a partner, at a distance of 5 m, one with a handball and one with a volleyball. Balls were thrown between partners for catching at the same time alternating with the left and right hands. At station 5, participants faced a partner at a distance of 5 m, one holding a volleyball in their hands and the other with a soccer ball at their feet. The balls were passed between partners simultaneously alternating hands/feet with each pass.

### 2.5. Breakfast

The breakfast provided to participants in the B-S and B-E groups employed common foods which were easy to consume by local children and adolescents. It consisted of a sandwich made from two slices of Gardenia Nutri multi-grain bread (estimated GI = 62) and 20 g of Nutella Ferrero hazelnut spread with cocoa (GI = 33), and a 200 mL packet of Nestlé Milo (GI = 35)—a chocolate malt powder drink popular in South-East Asia [29]. The breakfast provided 382 kcal (1.6 MJ) of energy and had 52.7 g of carbohydrate, 12.4 g of fat and 14.3 g of protein. The total meal GI was calculated based on an established formula [30]. The GI of each individual food was multiplied by the grams of available carbohydrate in the food, the total summed for all three foods in the meal and then divided by the total available carbohydrate in the meal. The calculated total meal GI was 46.

### 2.6. Blood Glucose

Blood glucose was measured from fingertip blood samples at the start (08:40 a.m., pre-test_1_) and end of the morning (12:30 p.m., post-test_4_). At both time points, a 30 μL sample of whole blood was collected into a capillary micro-tube. Samples were immediately dispensed onto a reagent strip and inserted into a dry-chemistry analyzer (Reflotron Plus, Roche Diagnostics, Mannheim, Germany). The analyzer was cleaned and calibrated according to the manufacturer’s specifications (Reflotron Clean & Check, Roche Diagnostics, Mannheim, Germany) and accuracy and precision was monitored using quality control sera (Reflotron Precinorm U Control, Roche Diagnostics, Mannheim, Germany).

### 2.7. Academic Tests

Academic performance was assessed with the Wechsler Individual Achievement Test, Third Edition (WIAT-III) [31] at the first and final test-series of the morning. The WIAT-III is a comprehensive, self-administered achievement test designed for children and adolescents. Subtests of the WIAT-III were used to determine mathematic and oral word fluency under time-limited conditions of 60 s. For mathematic fluency, participants were given 60 questions to complete which assessed the speed and accuracy of simple addition, subtraction, and multiplication calculations. They were instructed to answer as many questions as possible in the time provided. The total number of correct questions answered (score) and the number of correct questions answered each second (speed/computational time) was calculated. For oral word fluency, participants completed tests that measured the efficiency of their word retrieval to a given category, with 60 s provided for recall for each category. Participants were encouraged to provide as many words as possible. Proper nouns were excluded and the total number of words provided and time taken per word retrieval calculated. For both mathematic and oral word fluency, two versions of each test were provided to participants in a random manner at the start and end of the morning.

### 2.8. Cognitive Tests

Three tests to measure different aspects of cognitive function were provided using Inquisit software (Millisecond Software 2014, Inquisit 4, Seattle, WA, USA) in the same order at each test-series assessment point over the morning. Selective attention and inhibition were assessed using the Stroop Color-Word Test. In this computer test, cognitive interference was enhanced by asking participants to identify the color of ink in which a color name was written and to disregard the color name itself. The test consisted of three different conditions: (i) congruent trials; (ii) incongruent trials; and (iii) control trials [32]. The Digit-Span Test was employed as a working memory task in which participants read a series of single digits and were asked to repeat those digits in either a forward or backward sequence [33]. The number of digits committed to memory increased with the length of the test. Once a participant was unable to repeat back the digits at a given difficulty level on two occasions, the task was terminated. Attention was assessed using the Go/No-Go Task [34]. During this task green and black rectangles were presented on a computer screen as a continuous stimulus with participants having to react and perform a binary decision for each stimulus with a green rectangle requiring a motor response (Go) and presentation of a black rectangle requiring participants to withhold their response (No-Go). Accuracy and reaction time were measured for each event.

### 2.9. Hunger, Satiety, Fullness and Appetite

Hunger, satiety, fullness and appetite were measured by participants marking 100 mm visual analogue scales for each variable [35]. These measures were made before and after each academic and cognitive test-series.

### 2.10. Motivation, Arousal, Feeling and Mental Effort

A participant’s motivation was measured before each test-series and evaluated using a 10-point single-item scale ranging from 1 (“not interested at all”) to 10 (“very interested”). Before and after each test-series arousal and feeling were measured by the Felt Arousal Scale [36] and a Feeling Scale [37], respectively. The Felt Arousal Scale is a 6-point single-item scale ranging from 1 (“low arousal”) to 6 (“high arousal”). The Feeling Scale is an 11-point bipolar scale of pleasure and displeasure that ranges from −5 (“very bad”) to +5 (“very good”). Immediately after each test-series, the associated mental effort was measured using the Rating Scale for Mental Effort [38]. Participants were instructed to rate the level of psychological effort using a scale ranging from 0 to 150, with increments of 10 shown on the left edge of the scale and nine category anchors on the right edge of the scale. These include “no effort at all” (at 3 on the scale), “a fair amount of effort” (at 58 on the scale), and “extreme effort” (at 114 on the scale).

### 2.11. Statistical Analysis

An independent *t*-test was used to examine differences in heart rate and RPE during exercise. For blood glucose, and the cognitive, academic and subjective data over the morning, linear mixed modelling was conducted with hierarchical linear and nonlinear modelling (HLM) software (Version 7.01) [39] based on full maximum likelihood estimation. Linear mixed modeling was employed due to the hierarchical structure of the data. That is, each time measurement of a variable (Level 1) was nested within each individual (Level 2). To check if linear mixed modeling was suitable for the present data, an intraclass correlation coefficient (ICC) was calculated for each of the outcome variables based on an unconditional model in which no predictors were included. After confirming the suitability of multilevel modeling as a method of analysis for the current data, linear mixed modeling was conducted on each of the outcome variables (blood glucose, academic tests, cognitive tests, subjective rating variables). For the variables which were measured more than twice, time and time^2^ were entered as within-individual (Level 1) predictors in the model. The combined intervention (B-E) was used as a reference group and compared with the control group (F-S) and the two individual interventions (B-S and F-E). This was in order to examine the improvements brought about by the combined intervention over the other interventions. When time, time^2^ and group were entered into the equation, they were dummy coded (time: 0–2; time^2^: 0–4; group: B-E = 0, F-S = 1, B-S = 1, F-E = 1) and therefore they were not centered. To statistically control the effect of psychological variables on the outcome variables of academic and cognitive tests, feeling and motivation scores were centered and included into the model as between-individual (Level 2) predictors. For all analyses, significance was set as *p* < 0.05.

## 3. Results

### 3.1. Participants

Written informed parental consent and adolescent assent was obtained from 101 volunteers. However, 18 participants failed to complete main trials after informed consent and screening, primarily for reasons of time, thus leaving 83 individuals who were randomized to the four groups. One participant was excluded from the data analysis because they failed to complete several self-report measures (feeling, motivation) which were included as between-individual (Level 2) variables in the data analysis; HLM software does not allow missing values at higher levels. Thus, 82 participants (64 female) who had no missing values at Level 2 were included. Some descriptive characteristics of these individuals are provided in Table 1.

### 3.2. Exercise

Heart rate during the 8 min of treadmill running differed between the F-E and B-E groups (t = −2.239, *p* = 0.033) and was 150 (17) bpm and 161 (11) bpm, respectively. However, RPE at the end of the running was similar (t = −0.521, *p* = 0.605) at 13.2 (1.5) and 13.4 (1.5) (“somewhat hard”), respectively. During coordinative exercise, the heart rate for F-E and B-E groups again differed (t = −2.661, *p* = 0.012) and were 125 (17) bpm and 139 (13) bpm, respectively, whilst RPE at the end of these coordinative exercises was similar (t = −0.672, *p* = 0.506), rated as 10.9 (2.2) and 11.4 (2.1) (approximately “light”), respectively.

### 3.3. ICCs

ICCs of the outcome variables ranged from 15.1% (blood glucose) to 75.9% (mental effort), showing that variance of the outcome variables occurred at the between-individual level. When the ICC is greater than 10% of the total variance in the outcome variable, the multilevel structure of data should be considered [40]. These high ICCs supported the rationale for using linear mixed modeling as an appropriate form of statistical analysis for the present data.

### 3.4. Blood Glucose

Fingertip blood glucose was obtained from 57 participants (F-S: *n* = 16; B-S: *n* = 16; F-E: *n* = 13; B-E: *n* = 12). The remaining 25 individuals expressed discomfort at the time of the first sample procedure and so no samples were taken. In the fasted state, mean blood glucose concentrations ranged between 4.6 and 4.8 mmol∙L^−1^ in all groups and the concentrations in the F-S, B-S and F-E groups were similar to the B-E group (all *p* > 0.05). Blood glucose increased in the B-E group and the B-S group over the morning, and the change in the B-S group (β = −0.14, *p* = 0.558) was similar to B-E (Figure 2). Compared with the B-E group, blood glucose fell over the morning in the F-S and F-E groups (F-S: β_11_ = −1.45, *p* < 0.001; F-E: β_13_ = −1.28, *p* < 0.001). Mean blood glucose was ≥ 1.19 mmol∙L^−1^ greater in both breakfast groups at the end of the morning than the two fasted groups.

### 3.5. Academic Tests

Mathematics test scores in F-S and B-S groups were similar to the B-E group for the baseline measure at the start of the morning (both *p* > 0.05) but were higher in the F-E than B-E group (*p* = 0.044). Mean test score increased in all groups over the course of the morning (Figure 3a). The increase was greater in the B-E than F-S group (β_11_ = −3.30, *p* = 0.014) but B-E did not differ from the B-S (β_12_ = −1.68, *p* = 0.253) or F-E (β_13_ = −2.81, *p* = 0.078) groups. Computational time (mathematics speed) for B-E was similar to the other three groups at the start of the morning (all *p* > 0.05). Computational time improved over the morning in all groups; in B-E to a greater extent than F-S (β_11_ = −0.070, *p* = 0.004) and F-E (β_13_ = −0.057, *p* = 0.041) but not B-S (β_12_ = −0.042, *p* = 0.101) (Figure 3b). Percentage changes in both mathematics test score (F-S: 7.2%; B-S: 11.0%; F-E: 10.4%; B-E: 15.2%; Figure 4a) and computational time (F-S: 7.4%; B-S: 11.1%; F-E: 11.0%; B-E: 16.7%; Figure 4b) over the morning are shown in Figure 4.

For oral word fluency, there were no differences in the total number of valid words retrieved or speed (time taken) of word retrieval (data not shown) in the F-S, B-S and F-E groups at the start of the morning (all *p* > 0.05) or the change over the morning (all *p* > 0.05) compared with the B-E group.

### 3.6. Cognitive Tests

Outcomes from the Stroop Color-Word Test are in Table 2. There were no differences in reaction time or response accuracy between the B-E and other groups at the start of the morning for congruent, incongruent or control trials. However, the improvement in reaction time for congruent, incongruent and control trials in the B-E group from the start to middle of the morning was greater than that of the F-S group (congruent: β_11_ = 115.67, *p* = 0.024; incongruent: β_11_ = 158.75, *p* = 0.046; control: β_11_ = 104.20, *p* = 0.036) but not different from F-E (congruent: β_13_ = 24.39, *p* = 0.664; incongruent: β_13_ = −69.52, *p* = 0.473; control: β_13_ = 73.00, *p* = 0.220) or B-S (congruent: β_12_ = 111.81, *p* = 0.072; incongruent: β_12_ = 69.10, *p* = 0.404; control: β_12_ = 111.30, *p* = 0.104) groups. The improvement in reaction time in B-E did not result in improved response accuracy from the start to middle of the morning compared with F-S (congruent: β_11_ = −2.08, *p* = 0.284; incongruent: β_11_ = −0.11, *p* = 0.976; control: β_11_ = 2.01, *p* = 0.367), B-S (congruent: β_12_ = −1.92, *p* = 0.414; incongruent: β_12_ = 3.89, *p* = 0.257; control: β_12_ = −0.80, *p* = 0.684) or F-E (congruent: β_13_ = −2.58, *p* = 0.201; incongruent: β_13_ = 0.33, *p* = 0.931; control: β_13_ = 0.20, *p* = 0.912) (Table 2). From the middle to end of the morning the improvement in reaction time was greater for the F-S (congruent: β_21_ = −48.01, *p* = 0.033) and B-S (congruent: β_22_ = −55.54, *p* = 0.029) than B-E group for the congruent trial only (incongruent and control trials all *p* > 0.05). There was no difference between F-E and B-E in reaction time for any of the trials over the same time period (all *p* > 0.05). There were also no differences in response accuracy among groups for any of the trials between the middle and end of the morning (all *p* > 0.05). Interference scores (i.e., incongruent − congruent, control, or (congruent + control)/2) were also examined for reaction time and response accuracy. However, F-S, B-S and F-E groups’ interference scores were not significantly different compared with the B-E group over the morning (all *p* > 0.05).

Outcomes from the Digit-Span and Go/No-Go Task are presented in Table 3. For the Digit-Span Test, no differences were present between F-S, B-S and F-E with the B-E group at the start of the morning (all *p* > 0.05) and the maximum number of digits memorized in the forward and backwards direction was similar with B-E in all groups across the morning (all *p* > 0.05). In the Go/No-Go Task no differences existed in the overall mean for correct reaction times or the overall error rate between B-E and the other groups at the start of the morning (all *p* > 0.05). Reaction time improved to a greater extent in B-E from the start to mid-morning compared with F-E (β_13_ = 33.99, *p* = 0.019) but this finding for F-E was reversed from the middle to end of the morning (β_23_ = −15.29, *p* = 0.021). There was no difference in reaction time improvements between B-E and the other two groups across the morning (all *p* > 0.05). There were no differences in Go/No-Go Task error rates among groups over the morning (all *p* > 0.05).

### 3.7. Hunger, Satiety, Fullness and Appetite

The two fasting groups (F-S, F-E) reported greater measures of hunger, and less satiety and fullness at the start of the morning compared with B-E (all *p* < 0.05), and for the F-S group a greater appetite (*p* < 0.05) as well (Appendix A). There was no difference in these measures between B-E and B-S (all *p* > 0.05). Hunger and appetite increased to a greater extent in the two fasted groups over the morning compared with B-E (all *p* < 0.05) but with no difference between B-E and B-S (all *p* > 0.05).

### 3.8. Motivation, Arousal, Feeling and Mental Effort

Motivation and arousal were similar at the start of the morning and across the morning in B-E compared with the other three groups (Appendix B). Mental effort in B-E was also similar at the start of the morning compared with the other three groups. Mental effort perceived by the B-E group significantly dropped from the start to middle morning compared with the F-E group (β_13_ = 16.63, *p* = 0.042) but the change was thereafter similar until the end of morning among all groups (all *p* > 0.05). Whilst feeling did not differ in B-E with the other groups at the start of the morning, feeling (good/pleasure) improved in B-E from the start to mid-morning compared with the other three groups (B-S: β_12_ = −0.59, *p* = 0.040; F-S: β_11_ = −0.59, *p* = 0.038; F-E: β_13_ = −0.64, *p* = 0.036) but were again similar until morning end (all *p* > 0.05).

## 4. Discussion

In this experimental study, adolescents who consumed a low GI breakfast and performed 30 min exercise showed greater improvements in their mathematics score and computation time on a simple arithmetic test compared with those who remained fasted and sedentary over a morning. In addition, adolescents in the combined intervention had temporary mid-morning improvements in reaction times for congruent, incongruent and control trials during an interference (Stroop Color-Word) test and in an attention (Go/No-Go) task. However, these faster reactions did not translate into improved task accuracy and the improvement in reaction time was obviated by morning end. This investigation builds on the work of one previously published study examining acute differences in cognitive outcomes in youth given a high- or low-GI breakfast in combination with exercise [25]. We believe that our study is the first to demonstrate an academic performance outcome after combining breakfast with exercise using an acute experimental design. Nevertheless, it is important to quantify that we did not see any clear effect of the combined intervention over individual interventions, despite increases for mathematical score and computation time in the combined group being greater than those observed in the breakfast or exercise groups alone.

Breakfast intake in children has declined in many countries. In one large multi-country comparison involving >190,000 adolescents aged 11–15 years a frequency of daily breakfast consumption of >70% was found in only 4 of 41 countries [41]. Other cross-sectional studies also indicate that breakfast skipping ranges from 10% to 30% among youth [42,43]. Childhood physical inactivity is a growing issue in many countries and despite consensus that time spent in physical activity does not hinder academic performance, the perception persists that time spent on non-academic pursuits negatively impacts schoolwork [44]. Our data showing improved mathematical accuracy and increased processing speed with the combined intervention may therefore be important in overcoming such beliefs. Moreover, it could encourage parents, educators and policy makers to implement healthy lifestyle programs for better classroom outcomes.

Previous evidence demonstrates the benefits of both breakfast and exercise on improving academic outcomes and cognitive function in adolescents but we are aware of only one investigation examining their combined effects on cognition alone [25]. Similar to that previous study, we used a Stroop Color-Word Test as a measure of selective attention and inhibition and also found improved response times but not accuracy in the combined breakfast and exercise condition compared with adolescents who were fasted and sedentary. The faster response times observed were for all Stroop Test trials, including during interference in the incongruent trials, demonstrating that even complex responses were improved. These observations agreed with those of our test of attentional control, the Go/No-Go Task, where again response time was faster but accuracy was not improved. We did not see any effect on our test of working memory (Digit-Span Test). Our findings are an important confirmation of the previous work [25] and reassuringly similar given that our participants were in late adolescence when growth and brain development are more similar to that of adulthood.

That the cognitive improvement was only for reaction times may not be surprising. Literature on the effects of breakfast and exercise on cognitive tasks is inconsistent. For example, breakfast can have domain-specific effects on areas of cognition such as memory but this is influenced by factors such as the cognitive load imposed and even gender [1]. Acute exercise may also impact different cognitive domains differently. One suggestion is that exercise induces increases of neurotransmitters (norepinephrine and dopamine) that have a positive effect on processing speeds but which could induce neural noise in the brain hindering working memory tasks [45]. Certainly, the influence on reaction times is further supported by one meta-analysis which showed that moderate intensity exercise, such as that used here, results in increased processing speeds [45]. In the same analysis, the authors suggest that effects on task accuracy are limited because the tests used are not sufficiently complex to measure exercise-induced changes in accuracy performance [45]. Future studies should consider including more complex tasks to evaluate this.

An important point to highlight was that the improvement in response time was at test-series_2_ immediately after the combined intervention. The improvement did not last until test-series_3_ when reaction times for both the Stroop Color-Word Test and Go/No-Go Task were similar in B-E with all other conditions. In part, this could be from progressive waning of the acute influences of breakfast and exercise session on these cognitive processes. Transient effects on cognition with breakfast intake (see breakfast mechanisms below) are known but the duration of any effect varies among studies. Similarly, meta-analysis demonstrates that post-exercise cognitive task performance often improves but the size of any effect on performance diminishes over time [11]. The neurotransmitter catecholamine, noradrenaline, only has a half-life in the peripheral circulation of 3 min and it has been argued that this may be one explanation [45]. However, this explanation has been challenged as the half-life of brain (central) catecholamine concentrations is considerably longer at 8–12 h [46]. An alternative explanation is the influence of exercise on the protein brain-derived neurotrophic factor (BDNF) which acts as a stimulus for signaling pathways involved in neurogenesis and neuroplasticity. The effect of BDNF on downstream pathways takes time and so is most likely observed post-exercise [45]. However, BDNF increases tend to be greater for higher intensity exercise protocols and heart rate and RPE data from our exercise indicate that these were of a more moderate intensity [11]. Thus, any effect on BDNF may have been limited.

We believe the present work is the first to show the combined benefit of breakfast and exercise on performance of simple mathematical calculation accuracy and processing speed—an important demonstration from an education perspective. The fact that this improvement occurred with a single morning’s intervention affirms previous literature that some of the improvement elicited from breakfast and exercise is from the acute effects rather than long-term changes in nutritional status or physical fitness [1,2,3,4,5,11,12,13,14]. Our study design offers a novel perspective to previous work by providing a direct comparison with a morning where individuals were fasted and sedentary as well as the individual interventions themselves. One observation is that the period of improvement with the mathematics test (test-series_3_) did not correspond with that of the cognitive tests (test-series_2_). Decreased task sensitivity with repeated testing for the cognitive tests may explain this but certainly the improvement in mathematics accuracy and processing speed remained within the time frame previously reported for breakfast or exercise [1,11,46]. Despite the important demonstration of increased mathematics accuracy and processing speed with the combined intervention, we are cautious about the extent of the improvement achieved. Although the mean percentage change in the B-E group for mathematics score and computational time was more than double the percentage observed in the F-S group, the raw scores at the end of the morning did not differ between groups suggesting the improvement was from different starting points. Nevertheless, no significant difference among groups at the start of the morning was seen and we do not believe that the improved scores in the B-E group are simply an artefact as similar improvements were not observed in the F-E or B-S groups. Similarly, we draw attention to the extent of the absolute mathematics improvements in the combined group over the morning which were smaller than that of the differences among groups at baseline. Thus, the overall effect of the intervention is limited. To some extent this is reassuring as no real improvement in mathematics ability would be expected in response to acute breakfast and exercise over a morning but rather the improvements reflect the limited effects of the intervention themselves.

A systematic review including 34 studies supports the assertion that acute breakfast intake improves cognition in both in well-nourished children and those considered nutritionally at risk. Breakfast has a transient beneficial effect on cognition for up to 4 h post-ingestion, particularly for tasks related to attention, executive function and memory [1]; a time period within that of the present study. Several mechanisms are postulated to explain the effect of breakfast intake on academic and cognitive performance including satiation and hunger alleviation [4,7], improved fuel and nutrient provision for the central nervous system [1,6], and an increase in the synthesis or levels of brain neurotransmitters [1,6]. Evidence indicates that one or several of these mechanisms were present in our study including lower subjective hunger and appetite, greater subjective fullness and satiety, and higher glucose concentrations. Some of this may stem from the low GI meal we employed rather than a high GI meal [2,3,4,6,8]. One note of caution here, however, relates to the subjective measures of hunger in our study, which although improved on the B-E compared with fasted groups over the morning, differed at the outset. We told participants at the start of the morning upon reporting to the laboratory which group they were in instead of blinding them until the treatment was given. In retrospect, this may have inadvertently biased their ratings on these measures. Behavioral explanations are another alternative possibility for the effects of breakfast on cognitive performance as short-term breakfast consumption can heighten subjective feelings of alertness and motivation to concentrate and learn [1]. However, we did not see greater motivation or arousal with B-E compared with fasting. Instead, feeling (good/pleasure) improved in B-E compared with the other three groups and mental effort in B-E dropped compared with the F-E group from the start to middle morning. These results suggest that the combination of breakfast and exercise is beneficial to enhance good feeling and make adolescents feel ready to engage in a task. Finally, there is recent evidence that intake of cocoa/cocoa products, as a source of dietary polyphenols (primarily flavonols), can acutely improve cognitive abilities in young adults and children [47]. This includes in areas such as attention, working memory and processing speed with improved cerebral blood flow or cerebral blood oxygenation postulated as potential mechanisms for improvement. Cocoa was present in both the Milo (7%) and hazelnut spread (7.4%) used in the breakfast in the present investigation and may have contributed to the acute improvements in cognition observed. The extent to which this occurred is speculative without a non-cocoa control breakfast for comparison. Moreover, other studies showed effects of low GI breakfasts, with or without the presence of exercise, on similar cognitive outcomes [8,25]. Nevertheless, the addition of cocoa products to potentiate the effects of breakfast on cognition in adolescents is a potentially exciting area for future research.

The exercise protocol used was a combination of a short period of aerobic exercise and bilateral coordinative ball exercises for several reasons. Firstly, current evidence suggests a relationship between both cardiorespiratory fitness and aerobic physical activity with performance on cognitive tests and academic achievement tests [17,21]. Secondly, physical education lessons in many countries, such as Singapore, emphasize a range of activities that combine coordinative exercise and physical workout and our protocol part-imitated both demands. Finally, the bilateral exercises employed emphasize the ability to balance, react, adjust and differentiate to different stimuli within a short time. These exercises were observed to elicit greater post-exercise increases in attention and concentration for adolescents, compared with their counterparts instructed to undertake moderate intensity exercise without a specific coordinative emphasis in a physical education class [24]. It is postulated that bilateral exercises activate the cerebellum and areas of the prefrontal lobe related to mathematics processing and reading comprehension in children to a greater extent than exercise without the same coordinative emphasis [24]. The improvements observed with the combined intervention confirm this previous observation that this type of physical activity may have a role in eliciting short-term improvements in cognition and academic performance. Importantly, the bilateral coordinative exercises in the intervention were rated as “*light*” (RPE of 11) by the participants suggesting that exercise does not need to be of a more vigorous nature to achieve short-term improvements in these areas.

There are several limitations to the present study. The parallel group study design means that individual differences in the rate of learning may have affected our results. However, the randomization process should have limited this impact and potential issues with learning and ceiling effects on tests prevented use of a repeated measures design. Although the combined intervention improved mathematics computation time and reaction time on the Go/No-Go Task compared with exercise alone, most measures for the combined intervention did not differ from the breakfast or exercise given in isolation. Our initial power analysis suggested we would require 128 adolescents, or 32 per group, to elicit a worthwhile change from the control intervention. Thus, we cannot rule out the possibility of a type II error when comparing our combined intervention with the individual interventions. Given the consensus within the existing literature, however, it seems likely that both breakfast and exercise are beneficial for cognition and academic performance in adolescents. Nevertheless, the primary aim of the study was to examine the improvements brought about by the combined intervention, rather than the individual interventions themselves where a substantial body of literature already exists, which is why the B-E group was used as the reference. We did nevertheless conduct an exploratory analysis using F-S as the control intervention and this confirmed that the only difference observed for our major outcomes was in the B-E group compared with F-S, with no differences between F-S with the two individual interventions.

We did not record the usual breakfast consumption of the adolescents. It is possible that the acute effects of breakfast on cognitive and academic outcome measures vary depending on the extent to which individuals are habituated to breakfast intake. We also chose to give the exercise before the breakfast for reasons of digestive comfort. The sequence of the interventions may be important in eliciting the changes we observed and, in reality, children may eat breakfast before actively transporting themselves to school. Finally, practical reasons such as participants transporting to the laboratory and performing baseline measures meant that, in real-time, exercise (09:30 a.m.) and breakfast (10:00 a.m.) were relatively late in the morning. This extended the period of fasting and meant that breakfast feeding probably did not occur in conjunction with the normal daily routine, both of which could have affected our outcome measures.

## 5. Conclusions

Adolescents who consume a low GI breakfast and perform 30 min of aerobic and bilateral coordinative exercise can improve their mathematics score and computation time on simple arithmetic tests and their reaction times in more complex cognitive tasks compared with remaining sedentary and fasting over a morning. The extent of these improvements is limited but may nevertheless be important from the perspective of short-term academic performance in youth. Our findings lend support to the implementation of healthy lifestyle programs, such as school breakfast clubs and walking programs for active transport to schools, by policy makers, parents and educators.

## Figures and Tables

**Figure 1 nutrients-13-01278-f001:**
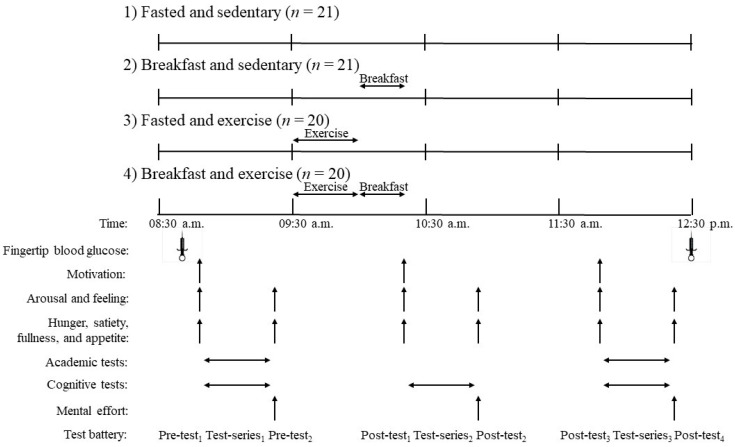
A schematic representation of the study protocol. Exercise involved 8 min of treadmill running at a speed equivalent to 60% of peak oxygen uptake followed by 20 min of bilateral coordinative ball exercises. Breakfast contained 382 kcal and consisted of a sandwich and a chocolate malt powder drink. Three cognitive tests (Stroop Color-Word Test, Digit-Span Test, and Go/No-Go Task) were conducted in consecutive order for each test-series. The Wechsler Individual Achievement Test (3rd Edition) was used for academic tests of mathematical and oral word fluency for test-series_1_ and test-series_3_.

**Figure 2 nutrients-13-01278-f002:**
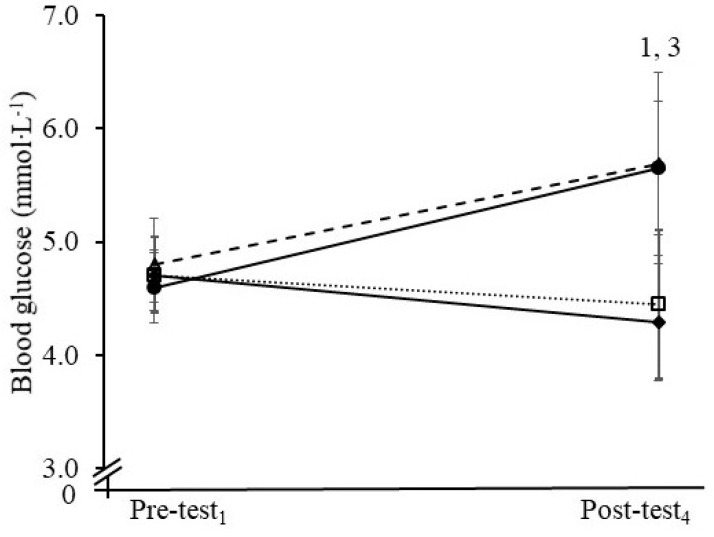
Blood glucose concentrations at the start (Pre-test_1_) and end of the morning (Post-test_4_) in the fasted and sedentary (F-S, ♦), breakfast and sedentary (B-S, ∆), fasted and exercise (F-E, □), and breakfast and exercise (B-E, ●) groups. ^1^ Change in B-E > than F-S over the morning, *p* < 0.05. ^3^ Change in B-E > than F-E over the morning, *p* < 0.05. Error bar: SD.

**Figure 3 nutrients-13-01278-f003:**
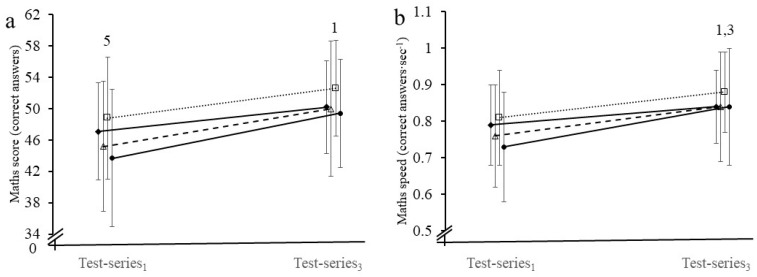
(**a**) Number of correct answers (mathematics score) and (**b**) correct answers per second (mathematics speed) in the fasted and sedentary (F-S, ♦), breakfast and sedentary (B-S, ∆), fasted and exercise (F-E, □), and breakfast and exercise (B-E, ●) groups in the 60 s arithmetic test during test-series_1_ and test-series_3_. ^1^ Change in B-E > than F-S over the test-series, *p* < 0.05. ^3^ Change in B-E > than F-E over the test-series, *p* < 0.05. ^5^ B-E different from F-E at test-series_1_, *p* < 0.05. Please note that there is no time lag in measurement among trials. Error bar: SD.

**Figure 4 nutrients-13-01278-f004:**
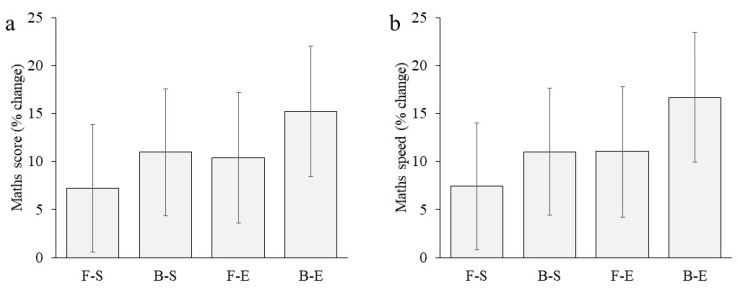
(**a**) Percentage change in mathematics score and (**b**) percentage change in mathematics speed in the fasted and sedentary (F-S), breakfast and sedentary (B-S), fasted and exercise (F-E), and breakfast and exercise (B-E) groups in the 60 s arithmetic test from test-series_1_ to test-series_3_. Error bars: 95% Confidence Intervals.

**Table 1 nutrients-13-01278-t001:** Physical characteristics of participants in the experimental trials.

Characteristic	F-S	B-S	F-E	B-E
*n*	21	21	20	20
Sex (M/F)	4/17	5/16	6/14	3/17
Age (years)	16.0 (1.3)	16.1 (0.8)	16.1 (0.9)	15.9 (1.2)
Body mass index (kg∙m^−2^)	20.4 (2.4)	20.8 (2.9)	20.9 (1.8)	20.0 (2.6)
Waist circumference (cm)	69.5 (4.8)	70.3 (7.5)	70.3 (4.7)	67.8 (7.2)
Peak oxygen uptake (mL∙kg^−1^∙min^−1^)	40.3 (5.8)	39.1 (5.1)	39.1 (5.4)	37.3 (3.8)

Data are means (standard deviations, SD). F-S: fasted and sedentary; B-S: breakfast and sedentary; F-E: fasted and exercise; B-E: breakfast and exercise.

**Table 2 nutrients-13-01278-t002:** Reaction times and number of correct answers for congruent, incongruent and control trials in the Stroop Color-Word Test.

	F-S (*n* = 21)	B-S (*n* = 21)	F-E (*n* = 20)	B-E (*n* = 20)
Congruent				
Reaction time (ms):				
Test-series_1_	750 (107)	800 (226)	830 (214)	787 (219)
Test-series_2_	697 (120)	735 (159)	713 (133)	665 (150) ^1^
Test-series_3_	649 (80)	661 (134)	653 (88)	656 (128) ^1,2^
Correct answers (no.):				
Test-series_1_	97 (4)	97 (4)	95 (5)	96 (4)
Test-series_2_	97 (3)	98 (3)	96 (5)	97 (2)
Test-series_3_	96 (4)	97 (3)	96 (5)	95 (4)
Incongruent				
Reaction time (ms):				
Test-series_1_	888 (170)	1003 (308)	1040 (302)	989 (336)
Test-series_2_	848 (210)	905 (256)	855 (196)	831 (207) ^1^
Test-series_3_	766 (113)	834 (235)	788 (161)	780 (174)
Correct answers (no.):				
Test-series_1_	88 (8)	89 (6)	91 (6)	91 (7)
Test-series_2_	91 (6)	94 (5)	93 (6)	94 (5)
Test-series_3_	92 (7)	93 (3)	91 (7)	93 (6)
Control				
Reaction time (ms):				
Test-series_1_	765 (120)	803 (206)	818 (216)	814 (215)
Test-series_2_	720 (147)	757 (154)	748 (134)	702 (147) ^1^
Test-series_3_	671 (103)	690 (144)	685 (94)	664 (111)
Correct answers (no.):				
Test-series_1_	95 (4)	97 (3)	96 (5)	97 (3)
Test-series_2_	97 (3)	96 (3)	96 (4)	97 (4)
Test-series_3_	96 (4)	96 (4)	94 (6)	95 (5)

Data are means (SD). F-S: fasted and sedentary; B-S: breakfast and sedentary; F-E: fasted and exercise; B-E: breakfast and exercise.^1^ Change in B-E ≠ change in F-S over test-series points, *p* < 0.05. ^2^ Change in B-E ≠ change in B-S over test-series points, *p* < 0.05.

**Table 3 nutrients-13-01278-t003:** Maximum number of digits memorized in a forward and backward sequence in the Digit-Span Test, and reaction times and number of errors committed in the Go/No-Go Task across the morning.

	F-S (*n* = 21)	B-S (*n* = 21)	F-E (*n* = 20)	B-E (*n* = 20)
Digit-Span Test				
Forward (no.):				
Test-series_1_	8.2 (1.7)	8.5 (1.1)	8.5 (1.0)	8.7 (1.3)
Test-series_2_	9.0 (1.7)	8.9 (0.9)	8.9 (1.2)	8.8 (0.9)
Test-series_3_	8.9 (1.3)	8.9 (1.2)	9.7 (1.4)	9.3 (0.9)
Backward (no.):				
Test-series_1_	7.7 (1.9)	7.8 (1.5)	7.8 (1.3)	8.0 (1.2)
Test-series_2_	8.2 (1.6)	8.1 (1.2)	8.3 (1.2)	8.7 (1.3)
Test-series_3_	8.7 (1.6)	8.3 (1.3)	8.9 (1.3)	8.9 (1.4)
Go/No-Go Task				
Reaction time (ms):				
Test-series_1_	357 (42)	354 (27)	352 (32)	345 (22)
Test-series_2_	345 (39)	356 (39)	362 (45)	335 (30) ^3^
Test-series_3_	344 (45)	345 (36)	347 (35)	332 (27) ^3^
	357 (42)	354 (27)	352 (32)	345 (22)
Errors (overall):				
Test-series_1_	0.012 (0.017)	0.023 (0.031)	0.036 (0.108)	0.007 (0.010)
Test-series_2_	0.010 (0.011)	0.022 (0.028)	0.034 (0.089)	0.011 (0.019)
Test-series_3_	0.015 (0.018)	0.013 (0.019)	0.015 (0.017)	0.012 (0.023)

Data are means (SD). F-S: fasted and sedentary; B-S: breakfast and sedentary; F-E: fasted and exercise; B-E: breakfast and exercise. ^3^ Change in B-E ≠ change in F-E over test-series points, *p* < 0.05.

## Data Availability

Data associated with this publication are deposited in the Data Repository of the National Institute of Education, Nanyang Technological University (Singapore) and will be made available upon publication of the manuscript: https://doi.org/10.25340/R4/UPNUXH (accessed on 10 January 2021).

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
