# Peer review of "Breakfast and Exercise Improve Academic and Cognitive Performance in Adolescents"

_nutrients, 2021, doi:10.3390/nu13041278_

Round 1

Reviewer 1 Report

In this study eighty-two adolescents (64 female) aged 14-19 years were admitted to a clinic for 4 hours following an overnight fast.  Subjects were randomized into 4 groups: a group who continued fasting and were sedentary (FS), a group who ate breakfast and were sedentary (BS), a group who continued fasting but completed  30-min of exercise (FE) and, a group who ate breakfast and completed 30-min exercise (BE).  A series of cognitive tests were performed immediately before the scheduled exercise/breakfast time immediately, immediately after breakfast, and approximately 1-hour later.  Incremental improvements in mathematics score were higher in BE than FS (15.2% v 6.7%;  p = 0.014) and computation time for correct answers (16.7% vs. 7.4%; p = 0.004, incremental change calculated as third test minus first test).  The BE group had faster reaction times for congruent, incongruent and control trials of the Stroop Color-Word Task compared with F-S mid-morning (all p < 0.05). The authors conclude that morning breakfast and exercise combine to improve short-term mathematical task performance and speed in adolescents.

Comments Major:

The results are reported as percent improvement between the first (pre exercise/breakfast, ~8:45 am) and third series of tests (post exercise breakfast, ~11:45 am), making it unclear how the data from the second series of cognitive tests (post exercise breakfast, ~9:45 am) were used.  Statistically, the data were analyzed using a linear mixed modeling approach on the measured changes, which indicated the improvement in academic tests (math) was BE were significant (p<0.05), and by comparing confidence intervals around the percent changes, which showed the improvements were not significant (overlapping confidence intervals).  Having an answer that depends on how the data are analyzed creates confusion.  Thus, the questions/comments below are largely related to the statistical-analysis/data-presentation.

  1. The authors state in the Methods section that: According to graphical inspection of the data, a quadratic trajectory trend was observed for the cognitive and subjective data that were measured more than twice. Therefore, time and time2 were entered as within-individual (Level 1) predictors in the model. Strictly speaking, this is a Result not a Method.  This analysis should be presented in the results section and the quadratic effect should be tested to determine if it is significant. 
  2. Ultimately, the authors rely on a mixed model approach to analyze their data. This is an appropriate approach.  However, the authors go into substantial detail on why the approach is preferable to an ANOVA approach, pointing out the main benefit being the ability to handle missing data.  The arguments are largely correct, but they are also well known and are unlikely to be of substantive interest to the reader.  The authors should consider removing these arguments and limiting the description of the statistical approach to what would be required for someone trained as a statistician to reproduce the analysis; i.e., just state the analysis used (mixed model) and the terms used in the model (subject, age, sex, treatment, etc.).

  1. The authors analyze the percent change in outcomes using what they refer to as a one-way ANOVA, which failed to detect any improvement among the 4 groups. They argue the ANOVA failed to detect differences in the percentage change among groups as it was calculated “using a single data point”.  This creates the impression that the ANOVA per se gives a different answer than what would be obtained using the mixed model approach, when in reality the two approaches, when given the same data will, in the absence of missing data which ANOVA cannot handle, yield identical p-values.  Here, the data changes from absolute changes to percent changes. Changing the data from absolute to percent will change p-values. Ultimately,  linear mixed model analysis on absolute scores should be thought of as more reliable.  In this reviewers opinion, the percent change analysis can be removed. 

  1. Although the reviewer does not believe the analysis on the percent change is needed, the reader is more likely to understand results that are presented as percent change.  Under normal conditions, no harm is done by assessing absolute changes for statistical significance, and reporting the accompanying changes in terms of percent.  However, this can be misleading when the denominator used to calculate percent change is significantly different in one group, as is the case in this study.  Essentially, the problem resides with the baseline score in the BE group being lower than the baseline score in the other groups (significantly lower compared to math scores obtained to FS; Figure 3a), which has the effect of inflating percent improvement (still another reason not to analyze percent changes).  In short, the significantly higher change in BE math score relative to FS math score has less to do with a higher post breakfast exercise score than it does with having a lower pre breakfast exercise score.  This makes the primary conclusion of the study – that breakfast/exercise improves math scores- somewhat suspect.

Based on a preponderance of data, including data from other studies, the answer is probably correct.  However, given the rather small effect observed here, the authors should consider adding a paragraph on the clinical significance of a single missed breakfast or morning exercise routine.  Here, it could be noted that the absolute change in the BE group is smaller than than the average difference observed between groups at baseline.  The conclusion should also be revised to reflect that the change observed is small. 

Comments Minor

  1. The sentence “At 1000 participants in the B-S and B-E groups were given a standardized breakfast to consume within 15 minutes… (ln 150) is in desperate need of a comma.  As written, a reader might infer that you gave B-S and B-E groups breakfast after you recruited the first 1000 participants.
  2. The Figures showing mean and SD would be greatly improved if the data in different groups were slightly shifted left, or right, to prevent the error bars from overlapping.

Reviewer 2 Report

The authors described the impact of breakfast and exercise on improve academic and cognitive performance in adolescents. The study was very well designed and all the individual stages in detail and accurately described. These are important results that can be used by different societies to implement healthy lifestyle programmes among young people. At the same time, it can be concluded that the results of the studies are quite obvious.

In terms of substance, the article submitted for review is not objectionable.

It begs only the question related to GI breakfast:

  • How was the GI of the individual products included in the breakfast determined?
  • The authors gave the GI value of the individual products included in the breakfast, while the GI of the whole breakfast was not specified. This should be supplemented, especially since the authors use the statement that breakfast was characterized by a low GI.
